# Population Based Average Parotid Gland Volume and Prevalence of Incidental Tumors in T1-MRI

**DOI:** 10.3390/healthcare10112310

**Published:** 2022-11-18

**Authors:** Tina Brzoska, Till Ittermann, Friedrich Ihler, Carmela Koch, Markus Blaurock, Robin Bülow, Henry Völzke, Chia-Jung Busch, Achim Georg Beule

**Affiliations:** 1Department of Otorhinolaryngology, Head and Neck Surgery, University Medicine Greifswald, 17475 Greifswald, Germany; 2Institute for Community Medicine, University Medicine Greifswald, 17475 Greifswald, Germany; 3Institute of Diagnostic Radiology and Neuroradiology, University Medicine Greifswald, 17475 Greifswald, Germany; 4Department of Otorhinolaryngology, Head and Neck Surgery, University Medicine Münster, 48149 Münster, Germany

**Keywords:** salivary gland volume, population-based imaging, whole-body magnetic resonance imaging, parotid gland tumors

## Abstract

Representative epidemiologic data on the average volume of the parotid gland in a large population-based MRI survey is non-existent. Within the Study of Health in Pomerania (SHIP), we examined the parotid gland in 1725 non-contrast MRI-scans in T1 weighted sequence of axial layers. Thus, a reliable standard operating procedure (Intraclass Correlation Coefficient > 0.8) could be established. In this study, we found an average, single sided parotid gland volume of 27.82 cm^3^ (95% confidence interval (CI) 27.15 to 28.50) in male and 21.60 cm^3^ (95% CI 21.16 to 22.05) in female subjects. We observed positive associations for age, body mass index (BMI), as well as male sex with parotid gland size in a multivariate model. The prevalence of incidental tumors within the parotid gland regardless of dignity was 3.94% in the Northeast German population, slightly higher than assumed. Further epidemiologic investigations regarding primary salivary gland diseases are necessary.

## 1. Introduction

### 1.1. Anatomy and Imaging of the Parotid Gland

Salivary glands of the head and neck ensure a sufficient saliva secretion for adequate food intake and processing, as well as the maintenance of a healthy oral flora. Impaired function results in xerostomia that increases the risk of oral infection and can substantially impact quality of life [1,2].

The parotid gland is an irregularly shaped organ in the head and neck region, where numerous anatomical structures are crowded into a dense space. Assessing parotid gland volume and distinguishing normal from pathologic conditions is challenging in an individual for lack of standard measurements. Sequential imaging by means of MRI is considered the gold standard for evaluating the parotid gland and its pathologies [3,4]. In clinical routine, an MRI of the parotid gland is the most important tool to assess the incidence and quality of parotid lesions even in the deep lobe [4,5]. Until now this imaging modality of the parotid gland has only been used in the clinical, not epidemiological context. For this reason, the incidence of benign and/or malignant parotid gland tumors are frequently reported, but valid figures of prevalence are missing.

Regarding the size of the parotid gland, general anatomy textbooks only describe the weight to be around 15–30 g [5,6,7,8,9]. Post-mortem or sialographic studies from the 1970s and 80s report mean volumes of 28–35 cm^3^ in small cohorts [10,11]. Until now only few studies with smaller cohorts, partly retrospectively and with different examination methods (nuclear medical examinations, MRI, CT scans or ultrasound) have found a mostly non-significant age and gender dependence of parotid volume [12,13,14,15]. Evaluating parotid gland volume in a population-based study is essential to establish reference values as well as to observe associations between age, sex, body morphology and common diseases such as type 2 diabetes mellitus (T2DM) or arterial hypertension (AH), which is a foundation for understanding physiologic and pathophysiologic changes of the parotid gland. Therefore, objectives of this study are to develop a reliable method to evaluate parotid gland volume and to determine an accurate prevalence of parotid gland lesions on MRI.

### 1.2. Study of Health in Pomerania: An Epidemiological Cohort Employing MRI Scans

The Study of Health in Pomerania (SHIP) represents a cross-sectional sample of the Pomeranian population in northeastern Germany. The overall aim of the SHIP is not only a long-term longitudinal observation of cohorts, but also the comparison of transversal cohorts, as well as the cross-regional comparability of epidemiological studies. It is the first population-based study to implement whole-body MRI [16]. Besides this fact, one of the first SHIP revelations was the pronounced prevalence of common diseases, such as type 2 diabetes mellitus and arterial hypertension in Pomerania compared to the national average. Among the adult population of Pomerania, the prevalence of diabetes mellitus was among the highest with 10.9 % (9.6–12.3%) compared to the national average of 8.2% (7.3–9.2%) [17]. To a large extent, diabetes mellitus is still under-diagnosed in the general German population [18]. Similarly, using the SHIP data, it was found that the prevalence of arterial hypertension was higher in northeastern Germany (60.1% in men, 38.5% in women) than in the southern German population (41.4% in men, 28.6% in women) or than the national average [19,20,21].

Higher rates of hyposalivation and microstructural changes of parotid parenchyma have been observed in diabetics. To a lesser extent, this is also the case in the presence of hypertension [2,22,23,24]. It is unclear whether there is a significant association between prevalence of these diseases and parotid gland size. Significant findings in MRI might be helpful for an early diagnosis or detecting progression of diabetes mellitus or arterial hypertension in the Pomeranian population.

In summary, we expected to be able to reliably measure parotid gland volume and describe the presence of parotid lesions using the MRI scans provided by the SHIP (ICC > 0.7 in inter- and intrarater reliability indicating a good reliability). Furthermore, our null hypothesis was that age, gender and common epidemiologic diseases have no impact on parotid volume as measured by MRI scan in an epidemiologic cohort.

## 2. Materials and Methods

### 2.1. Cohort and Selected Modalities of the Study of Health in Pommerania

We used data from two independent SHIP cohorts, SHIP-START-2 (the 10-year follow-up of the SHIP-START cohort) and SHIP-TREND-0 (the second SHIP cohort). Subjects in both cohorts were recruited and screened using the same protocols. Subjects were aged between 20 and 88 years and were selected from reporting data in a randomized, two-stage cluster method stratified by gender and age. Subjects with informed consent underwent a standardized 1.5 T whole-body MRI, except for those with contraindications for MRI (8.1%) [25]. For imaging of the parotid gland, an 8-channel neck coil was used. Thus, 3206 MRIs could be obtained for the present study.

Arterial hypertension was defined as increased systolic or diastolic blood pressure and/or the use of antihypertensive medication. Diabetes mellitus was defined based on self-reported diabetes or self-reported diabetes treatment and/or dietary treatment as well as age at time of diagnosis. BMI was obtained via standardized measurement of body weight and height and subsequent calculation.

### 2.2. Standard Operating Procedure

Prior to processing, each MRI was checked for complete capture of the parotid gland, especially for the cranial part of the parotid gland of both sides (in the axial plane the external ear canal served as an anatomic landmark) as well as the entire caudal pole. Criteria for exclusion were insufficient capture of the gland as well as imaging artefacts caused by adjacent implants or movement.

In axial T1-weighted images the tissue of the parotid gland was outlined and marked manually as region of interest (ROI) in each layer via the open source software OsiriX Dicom Viewer. Images had a slice thickness of 4 mm. Left and right gland were measured separately. Extra glandular tissue such as lymph nodes, blood vessels and extracapsular fat were excluded in each layer (Figure 1).

Tumors or salivary stones were marked separately and were excluded from parotid volume. Tumors were defined as lesions differing from the parotid tissue in signal intensity and being detectable in at least two plains and two layers. An example of an outlined tumor can be seen in Figure 2. After a ROI was established in each layer, these were summed up to estimate a volume.

### 2.3. Statistic Tools

All analyses were performed using Stata 16.1 (Stata Corporation, College Station, TX, USA). All significance tests were two sided. *p*-values < 0.05 were considered as statistically significant. Associations of age, sex, anthropometric characteristics, smoking status, diabetes mellitus, lipid markers, and arterial hypertension with parotid gland volume were analyzed in linear regression models.

To adjust for confounding factors, models were stratified by sex and adjusted for age smoking status, BMI, and smoking status (if not already exposure). To account for the drop out to the MRI examination, inverse probability weights were calculated. For the SHIP-START-2 population these weights were multiplied with the dropout weights describing the dropout of individuals from SHIP-START-0 to SHIP-START-2.

Minimum and maximum values of parotid parenchyma volume were evaluated separately for each side, and the mean, median and standard deviation (SD) were calculated. Pearson’s correlation coefficient r was calculated to compare the right and left glandular parenchyma volumes. Sex-specific differences between parotid volumes were compared using a Mann–Whitney U test.

In a pre-test setting interrater and intrarater reliability was measured, graphically presented in a Bland–Altman plot and intraclass correlation coefficients (ICC) were computed. ICC was considered to be ‘moderate’ when ranging between 0.5–0.75, ‘good’ when 0.75–0.9 and ‘excellent’ when >0.9 [26].

This project was approved of by the SHIP-MRI Commission according to the guidelines of the Institute of Community Medicine in Greifswald for the use of data.

## 3. Results

### 3.1. Analysis of Cohort

The primary cohorts SHIP-START and SHIP-TREND are epidemiologically representative [27]. In the course of MR image analysis, 91 subjects had to be excluded because of imaging artefacts. Due to technical reasons, i.e., limitations of the 8-channel neck coil with difficult positioning of individuals with a long neck or a stiff neck-shoulder configuration, 1390 subjects’ images did not contain the cranial part of either one or both parotid glands.

Finally, 1725 out of 3206 MRIs with complete capture of the parotid glands were analyzed. Recruiting study (SHIP-START-2 versus SHIP-TREND-0) showed no significant difference with respect to MRIs suitable for measurement. Exclusion form the study took place for technical reasons or effects on any of the factors analyzed below (see Table 1). For this reason and to increase statistical power, all subjects were analyzed sequentially as one MRI cohort (654 males, 1071 females; see Table 1). 188 diabetics (10.9%) and 837 hypertensives (49%) were identified, corresponding to the prevalence of the northeast German population [27]. The in- and excluded subjects differed with regard to age, sex, smoking status, BMI and the presence of diabetes mellitus. To account for the selection bias, all further multivariable analyses were weighted for dropout to the MRI measurements.

### 3.2. Analysis of Methods

Inter- and Intrarater Reliability, Intraclass Correlation Coefficients

After pre-test measurements of the volume of both parotid glands in 100 subjects performed by two different readers, we observed an interrater reliability for the right parotid gland with a mean bias of 7.94% and a consecutive 1.96-fold SD of 54.68% (left parotid gland: 16.73% with a 1.96-fold SD of 83.39%). The resulting Bland–Altman plot is illustrated as Figure 3. The measured values scatter evenly around the mean value, except for four outliers. Finally, an ICC of 0.72 for the right and 0.78 for the left parotid gland was calculated and considered ‘moderate’ and ‘good’ [26].

The intrarater reliability for 20 subjects’ parotid gland volumes measured at two different points of time showed a mean bias of −7.69% and a consecutive 1.96-fold SD of 16.22 for the right and −8.9% and 1.96-fold SD of 21.64% for the left parotid gland. This results in an ‘excellent’ ICC of 0.98 for the measurements of the right and 0.96 of the left gland [26]. Within the pre-test evaluation, none of the readers at any point of measurement found a parotid tumor, which results in an ICC of 1.

### 3.3. Volume of the Parotid Gland and Association to Sex, Age, BMI and Selected Comorbidities

In our study cohort the mean volume of right parotid glands was 24.32 cm^3^ (95 % confidence interval (CI) 23.90 to 24.74), and of left glands 23.50 cm^3^ (95 % CI 23.09 to 23.91) ranging from the smallest gland with 0.11 cm^3^ up to 61.84 cm^3^. Mean parotid volume was not significantly different between the two study cohorts (right: 24.3 cm^3^ in TREND vs. 24.5 cm^3^ in START, *p* = 0.605; left 23.4 cm^3^ in TREND vs. 23.8 cm^3^ in START, *p* = 0.275). Figure 4 shows a nearly normal distribution for different volumes of both sides.

There was no significant difference in volume between left and right gland within an individual (correlation coefficient r according to Pearson 0.89). All further analyses were performed with the subjects’ mean value from left and right gland.

#### 3.3.1. Associations of Sex and Age with Parotid Volume

Parotid gland volume was significantly higher in male subjects with a mean of 27.82 cm^3^ (95 % CI 27.15 to 28.50) than in female subjects with a mean of 21.60 cm^3^ (95 % CI 21.16 to 22.05) (*p* < 0.001). There was a significant effect of age on parotid volume in both women and men (*p* < 0.001) with increasing gland size in older age (see Figure 5).

#### 3.3.2. Associations of BMI with Parotid Volume

In both females and males, BMI was positively associated with parotid gland volume (*p* < 0.001). Association of BMI with parotid volume was stronger in males than in females as shown in Figure 6.

#### 3.3.3. Associations of Diabetes Mellitus and Arterial Hypertension with Parotid Volume

After adjustment for confounders, the group of diabetics has a tendency towards higher parotid volume with an adjusted mean of 28.8 cm^3^/22.4 cm^3^ (male/female) compared to non-diabetics with 27.6 cm^3^/21.5 cm^3^ (male/female). However, neither in males nor females the association of type 2 diabetes on parotid volume was statistically significant (men: *p* = 0.233; women: *p* = 0.113) (Table 2).

We observed a positive association between the presence of arterial hypertension and parotid gland volume in men (*p* = 0.027) but not in women (*p* = 0.056). (Figure 7, Table 2). In hypertensive males the adjusted mean parotid volume was 28.3 cm^3^, while in non-hypertensive males the adjusted mean volume was 27.0 cm^3^.

Overall, more significant associations were observed for the entire cohort than for either the male or female subgroups. In females, BMI, waist circumference, serum triglycerides levels, and plasmaHbA1c levels were positively associated with parotid gland volume. In the entire study population, we also found a positive association between systolic and diastolic blood pressure and parotid gland volume, which was not statistically significant when the male or female population was analyzed separately. We could not show any significant associations between smoking status and parotid volume. Again, when the results of the regression analysis were adjusted for the recruiting study (SHIP-START-2 versus SHIP-TREND-0), there were no significant changes in values or reported associations.

### 3.4. Prevalence of Tumors of the Parotid Gland

Within our cohort of 1725 subjects, we identified 75 parotid lesions in 68 subjects, which suggests a prevalence of 3.94%. The measured tumor volumes showed a broad dispersion between 0.6 cm^3^ and 7.5 cm^3^. Statistical details are shown in Table 3.

No sialoliths were identified, although large specimen could have been visible in a T2-weighed image as hypointense structures [28].

## 4. Discussion

### 4.1. Cohort and Imaging Modalities

Cohort

The SHIP cohorts START and TREND are a representative cross-section of the adult population of Northeast Germany. We were unable to demonstrate any difference between the subsamples from SHIP-START-2 and SHIP-TREND-0. Therefore, the combined results could be directly compared with other epidemiological studies, such as the southern German KORA study [17,27,29,30].

In comparison to the initial cohort, our study population was older, had a slightly higher BMI and a higher proportion of women (62%) than the original SHIP cohort (51.4%) or the population of Pomerania (50.9%) [27]. To counteract general bias, gland volume was reported differentiated by gender and linear regression analyses adjusted for study, sex, BMI and status of diabetes mellitus and arterial hypertension as well as inverse probability weighting was applied. The final cohort’s prevalence of diabetes mellitus (10.9%) and arterial hypertension (48.8%) lay within the range of the assumed prevalence in Pomerania [27]. Therefore, we still consider our values for parotid gland volume and incidence of tumor representative for the Pomeranian population [31,32].

Imaging Modalities and Standard Operating Procedure

MRI as gold standard for parotid gland imaging, has nearly no side effects for the individual. Due to technical deficiencies, our protocol showed a high number of excluded subjects. Nevertheless, it resulted in a sufficiently large cohort to develop a reliable SOP when subjects from SHIP-START-2 and SHIP-TREND-0 were analyzed as a combined cohort. In future, the MRI protocol should be adapted to have less drop outs.

The inter- and intrarater reliability and deriving ICC were obtained in a pretest setting and are a measure of the quality of the method used here. The resulting intrarater values and the excellent ICC (> 0.9) derived from them prove that the values are easily and accurately reproducible with our protocol. Our interrater measurements identified four extremely downward deviating values when re-measuring right parotid glands. Since the ICC is based on a univariate analysis of variance, it is susceptible to interference by extreme values [33]. Finally, our method’s interrater reliability (ICC > 0.7) was ‘moderate’ to ‘good’ [26].

The verification of the methodology by means of ICC has seldomly been described in previous studies. The good intra- and interrater reliability makes the method recommendable for further studies. Using this method, follow-up studies could assess volume changes over time. Applied by other population- and MRI-based studies, valuable information about regional differences could be obtained to evaluate the influence of genetics, dietary habits, and other common diseases.

Analyzing only non-contrast, T1-weighed MRI, no conclusions can be drawn regarding histological composition of the salivary parenchyma. For a future study, quantitative MRI with analysis of T1 and T2 relaxation times could differentiate between functional acini and fatty or connective tissue and establish standard values [34]. In the same manner, parotid lesions could not be analyzed for benign or malignant subtypes.

### 4.2. Comparison of Results in Different Studies

Using a variety of imaging modalities such as CT, PET-CT, and MRI, several recent studies investigated smaller subject populations (ranging from *n* = 16 to *n* = 240) in different clinical and experimental settings [12,13,14,15,34,35,36]. Their details are shown in Table 4. In these studies, the mean parotid volume data varied widely between 17 and 40 cm^3^. In some of these studies, a significant positive association between age, sex, and BMI and parotid volume was described, as in the present study.

Most of the reported volumes, both in MRI, CT and PET-CT scans, are comparable to our results. Not all, but many similar results were obtained for laterality, range of gland size and the positive association between male sex, age, and BMI with parotid gland volume. Alternate results could be due to selection bias, extreme outliers skewing small cohorts, clinical settings and/or a predominant proportion of male subjects. Not all studies excluded extra parenchymal tissue which would likewise result in slightly larger volume values [12,13,14]. Only Medbery et al. presented data on interrater reliability. None of the studies provided information on interrater reliability or derived ICCs.

In general, the different imaging modalities, inhomogeneous and epidemiologically non-representative cohorts, different slice thickness and methods of volume calculations, as well as lack of pre-test quality assessment in the previous studies makes a direct comparison with existing data difficult.

### 4.3. Associations between Sex, Age, BMI, Arterial Hypertension and Diabetes Mellitus and Gland Volume

#### 4.3.1. Influence of Sex on Parotid Volume

Similar sex-specific positive associations of male sex and volume for the large cephalic salivary glands were also found by Li, Heo, Niendorf, and Mahne et al. [12,13,15,37]. The reason for parotid gland volume differences by sex is not entirely clear. In other exocrine organs like the thyroid gland and the liver, associations between sex, age and BMI are observed likewise [38,39,40].

Histologically, independently of sex, the presence of estrogen and androgen receptors can be found in both acinar and ductal cells of salivary glands [41]. However, the associated physiological mechanisms and influence of sex hormones have not yet been elucidated. Sex-specific dimorphism of the large cephalic salivary glands has been demonstrated in the rat model but is not transferable to human salivary glands [42]. Hormone status, hormone replacement therapy, and age-related hormone changes could be influences. There are currently no disaggregated data on these details in relation to parotid volume, so no comparison can be made to existing literature. However, the sex differences give reason to do so in future epidemiological studies.

#### 4.3.2. Influence of Age on Parotid Volume

The values obtained in this study demonstrate an increase in the volume of the parotid gland with increasing age. Li, Ono and Heo et al. have observed the same positive association between age and gland volume [13,15,36]. For the submandibular salivary glands, this effect was also described by Niendorf and Saito et al. [34,37].

Histological studies highlighted the age-related, disproportionate increase of fibroblastic and adipocytic tissue and proportionally smaller decrease of functional acini within glandular parenchyma [43,44]. It is of note that factors such as a typical increase in BMI as well as elevated blood plasma glycerides might be the primary culprit for the age-related increase salivary gland size [45], despite the application of linear regression analysis with adjustments for these parameters.

Multiple research groups assessing salivary gland function by flow rate could not demonstrate an age-dependent effect of salivary flow rate, which supports the hypothesis of increased salivary reserve in the remaining acini in older age [43,44,46]. Glandular volume potentially increases slightly in aging, the loss of acinar cells balanced by an increase in connective and adipose tissue. Xerostomia or loss of parenchymal volume must be considered pathological.

#### 4.3.3. Influence of BMI on Parotid Volume

Similar significant positive correlations between BMI and gland size were described by research groups led by Dos Santos, Heo, and Ono et al. who also suggested that fat deposits are more likely to contribute to the total volume [15,35,36]. A positive association between BMI and parotid fat saturation in MRI has been observed [47,48]. Although the histological and volume changes of the parotid glands are not directly comparable to the submandibular glands, interesting parallels still exist. For example, by examining submandibular glands histologically post-mortem, Waterhouse et al. described an age- and BMI dependent quotient for functional tissue in relation to adipose and connective tissue, with adipose tissue increasing disproportionately with age and functional acinar cells decreasing between 25–50% at older ages. Interestingly, tissue ratios were not significantly different in obese subjects after regression analyses [44]. Physiologically, activation of salivary secretion is triggered by food intake. In the case of increased food consumption, the gland is stimulated accordingly. Sympathetic and parasympathetic nerve fibers are activated synergistically and lead to parasympathetic vasodilation among other things. The elevated perfusion meets the increased needs for water and substrate for higher secretory activity [49,50,51]. The increased blood flow and greater amount of intraglandular saliva, as well as the increased number of adipocytes, may be responsible for an overall greater salivary gland volume in individuals with a higher BMI. An increase in gland perfusion after stimulation can also be demonstrated MR-morphologically [50]. Future histological studies in combination with MRI could support this chain of argumentation.

#### 4.3.4. Discussion of Parotid Volumes in Relation to Selected Comorbidities

None of the studies mentioned above concerning the volume of the parotid gland commented on possible associations between diabetes mellitus and arterial hypertension and gland size. The observed positive association between the presence of arterial hypertension in men, elevated systolic and diastolic values, elevated HbA1c becomes apparent only in a large cohort. These associations have not been observed in living subjects up to now. Islas et al. found a positive association between the presence of diabetes and volume of large cephalic gland in a post mortem analysis [52].

In saliva samples from diabetics and hypertensive patients, the urea and total protein content is increased and the microalbumin content is decreased compared to non-diabetic and non-hypertensive patients to a lesser degree [23,24,53]. This suggests a change in salivary parenchyma composition in the presence of these systemic diseases. Since treatment status for diabetes or hypertension was not analyzed in this cohort, it is not possible to say whether antidiabetic or antihypertensive medications have an effect on gland size and whether untreated individuals might have larger salivary glands. However, because parotid volume is subject to greater inter-individual variation, these comparatively small differences in volume do not have any clinical relevance for the individual when looking for diagnostic tools.

### 4.4. Significance of the Detection of Parotid Lesions

This is the first study to state the prevalence of intraglandular parotid lesions, namely 3.94%. Salivary gland tumors are generally rare, but diverse and mostly benign. Malignant salivary gland tumors are rare with an incidence of 1–2 cases per 100,000 per year according to the German Cancer Registry [54]. Previous international studies describe an incidence of 2.5–5 of both benign and malignant tumors per 100,000 [55,56,57]. These data were obtained either from pathology institutes or from clinical centers specialized in parotid surgery and therefore able to recruit a sufficient number of cases for their studies. Due to a strong selection bias, these cohorts do not represent the average population.

This study demonstrates that parotid gland tumors are of a higher prevalence than previously assumed. MRI is the most reliable method to detect them [28]. None of the previous parotid volume studies had made comments on the prevalence or incidence of intraglandular parotid tumors. In most cases, pathologies within the head and neck region were exclusion criteria in these studies.

According to German guidelines, surgery is recommended for salivary gland tumors, the extent varying from enucleation of the tumor up to partial or total parotidectomy, the choice depends on tumor configuration, location and hints of malignancy [58]. In SHIP MRI follow-up examinations, the volume development of the detected tumors, resection rate and resulting entities should be examined up to give better guidelines for treatment.

### 4.5. Stength and Limitations

In this study, a reliable MRI-based method (ICC > 0.7 for inter- and intrarater reliability) for assessing parotid gland volume and tumor prevalence was developed for the first time. The measurements can be reproduced at any time by different investigators.

The present MRI dataset is the first to be obtained from large, population-based cohorts. The high number of excluded subjects, whose head and neck configuration caused technical difficulties, could lead to a bias that excludes larger subjects, leaving a tendency toward smaller mean values for the parotid gland overall. Future MRI protocols should be technically adapted to ensure complete imaging of the parotid gland. Since subjects of the final cohort were older, suffered more often from chronic diseases, and had higher BMI than the initial cohort, we did not only adjust for these confounders in regression analysis but also applied inverse probability weighting.

Further epidemiologic studies using similar methods could reveal regional differences or identify other associated variables such as differences in lifestyle, dietary habits, genetic predisposition, or other comorbidities.

It is the first publication to determine an accurate prevalence for incidental tumors of the parotid gland. Since only T1-weighed images have been analyzed, there is limited information about the entitiy of tumor or fat content of parotid tissue. Using glandular volume to draw conclusions about the presence of systemic disease is of little value in the individual because of the wide variation in volume. Therefore, the clinical utility of this protocol is limited.

## 5. Conclusions

With our measurement protocol, T1 MRI scans of the parotid gland provide a dependable method of assessing parotid gland volume. We provide reference data for parotid gland volume, which is significantly affected by sex, age and BMI. Evidence is emerging that diabetes mellitus and arterial hypertension also have an impact on gland volume. In addition, lesions can be reliably detected, so we recommend this protocol for future epidemiological studies.

## Figures and Tables

**Figure 1 healthcare-10-02310-f001:**
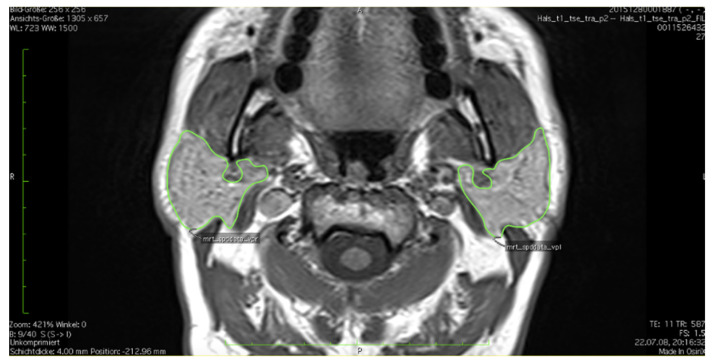
Example of the outlining of the parotid tissue in one single slice, axial T1-weighed MRI (Vv. retromandibulares are left out).

**Figure 2 healthcare-10-02310-f002:**
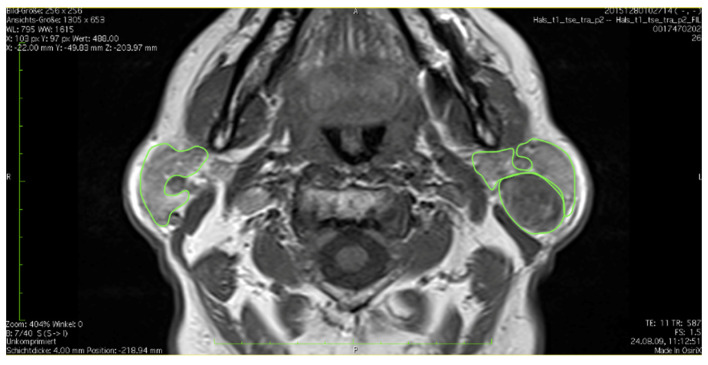
Example of the outlining of a lesion within the left parotid gland.

**Figure 3 healthcare-10-02310-f003:**
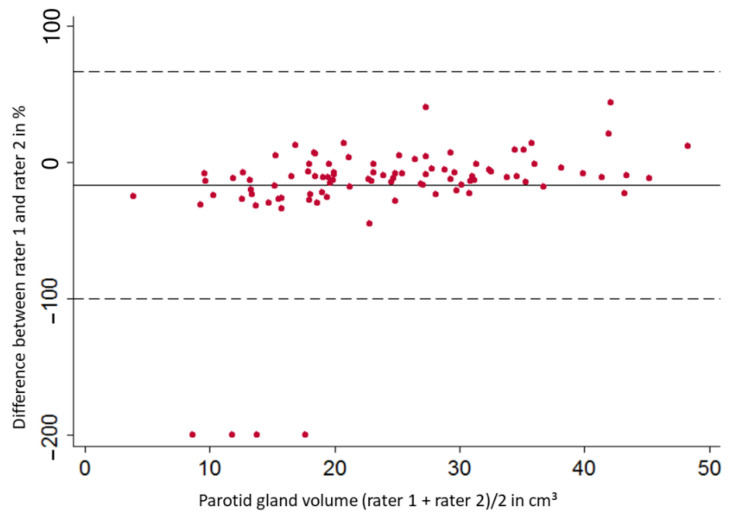
Bland-Altman plot for interrater reliability of right parotid glands.

**Figure 4 healthcare-10-02310-f004:**
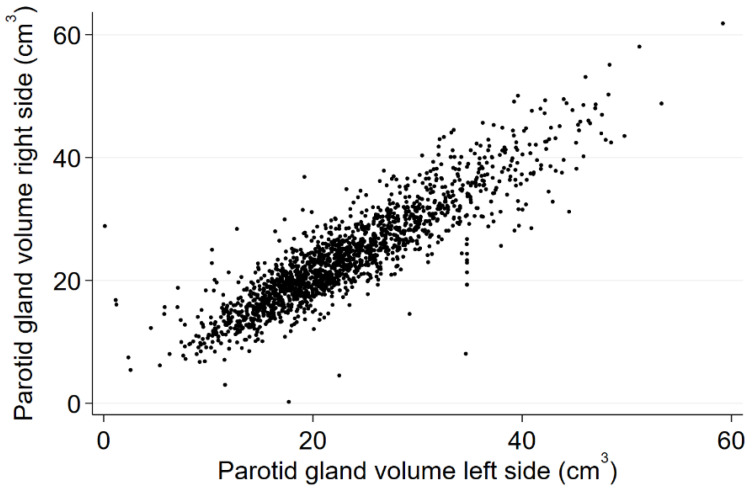
Distribution of volumes of left and right parotid glands.

**Figure 5 healthcare-10-02310-f005:**
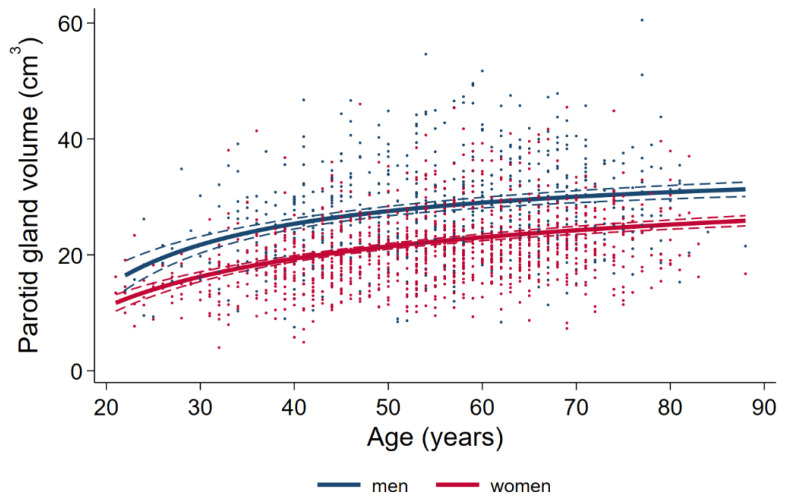
Association of sex and age with parotid gland volume differentiated by gender.

**Figure 6 healthcare-10-02310-f006:**
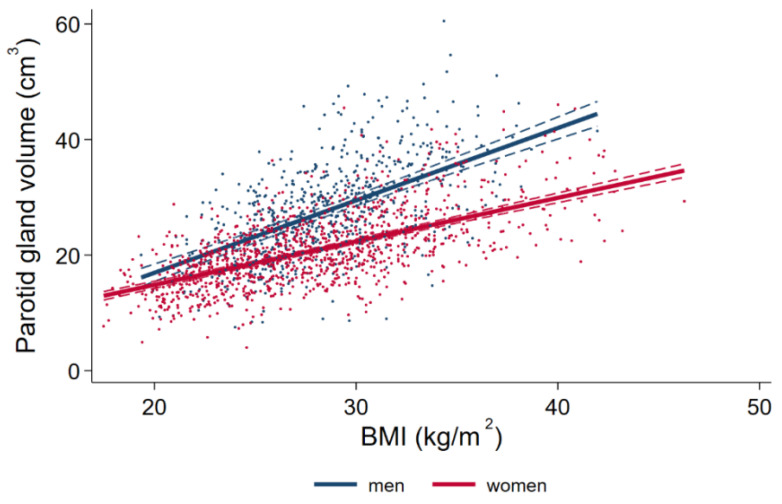
Association of BMI with parotid volume differentiated by gender.

**Figure 7 healthcare-10-02310-f007:**
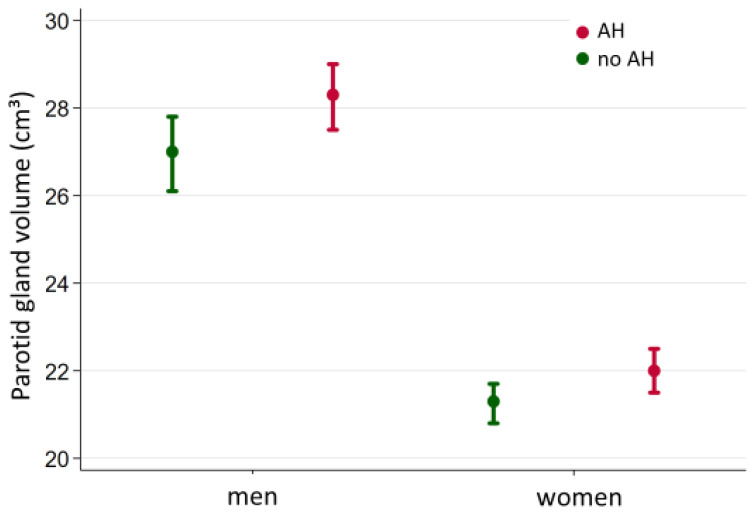
Volume differences in the presence of arterial hypertension differentiated by gender.

**Table 1 healthcare-10-02310-t001:** Analysis of excluded and included subjects.

	Excluded Subjects (*n* = 1481)	Included Subjects (*n* = 1725)	*p* *
Age; years	49 (40; 60)	56 (45; 65)	<0.001
Male; %	867 (60.5%)	654 (37.9%)	<0.001
Status of Smoking Never Formerly Presently	517 (35.0%) 577 (39.1%) 383 (25.9%)	742 (43.1%) 650 (37.8%) 330 (19.2%)	<0.001
Body Mass Index	26.8 (24.0; 29.8)	27.8 (24.8; 30.9)	<0.001
Body Weight; kg	80.7 (70.4; 92.3)	77.6 (67.9; 88.4)	
Body Height; cm	173 (167; 180)	167 (161; 173)	
Waist Circumference; cm	90 (80; 99)	90 (80; 100)	0.569
HDL-Cholesterol; mmol/L	1.39 (1.14; 1.67)	1.43 (1.21; 1.70)	0.010
LDL-Cholesterol; mmol/L	3.32 (2.71; 3.90)	3.40 (2.75; 4.04)	0.110
Triglycerides; mmol/L	1.31 (0.92; 2.01)	1.43 (0.98; 2.01)	<0.001
Glucose; mmol/L	5.3 (4.9; 5.8)	5.4 (5.0; 5.9)	0.142
HbA1c; %	5.2 (4.8; 5.6)	5.3 (5.0; 5.7)	<0.001
Systolic Blood Pressure; mmHg	128 (116; 139)	128 (115; 140)	0.833
Diastolic Blodd Pressure; mmHg	78 (71; 84)	78 (72; 85)	0.585
Arterial Hypertension	634 (42.9%)	837 (48.8%)	0.001
Type 2 Diabetes Mellitus	108 (7.3%)	188 (10.9%)	<0.001

Continuous data described by median, 25th and 75th percentiles; categorical data by number and proportion in %. * Wilcoxon-Test for continuous data, χ2-test for categorical data.

**Table 2 healthcare-10-02310-t002:** Overview on habitus, metabolic variables, smoking status and its associations to parotid gland volume in the final cohort.

	All Subjects (*n* = 1725)	Men (*n* = 654)	Women (*n* = 1071)
**Status of Smoking**			
**Formerly vs. Never**	0.60 (−0.08; 1.40)	0.57 (−0.85; 1.99)	0.70 (−0.15; 1.54)
**Presently vs. Never**	−0.05 (−0.97; 0.87)	−1.07 (−2.86; 0.72)	0.60 (−0.43; 1.63)
**Body Mass Index; kg/m^2^**	0.87 (0.81; 0.93) *	1.25 (1.11; 1.40) *	0.75 (0.69; 0.81) *
**Waist Circumference; cm**	0.36 (0.33; 0.38) *	0.43 (0.38; 0.48) *	0.32 (0.30; 0.35) *
**HDL Cholesterol; mmol/L**	−1.01 (−1.84; −0.18)	−0.77 (−2.53; 0.99)	−1.16 (−2.02;−0.31)
**LDL Cholesterol; mmol/L**	0.37 (0.08; 0.66) *	0.19 (−0.36; 0.75)	0.57 (0.25; 0.89) *
**Triglycerides; mmol/L**	0.70 (0.44; 0.97) *	0.55 (0.16; 0.94) *	0.83 (0.45; 1.21) *
**Type 2 Diabetes Mellitus**	1.27 (0.38; 2.16)	1.63 (−0.02; 3.28)	1.02 (0.04; 2.00)
**Glucose; mmol/L**	0.50 (0.30; 0.70) *	0.55 (0.20; 0.89) *	0.40 (0.17; 0.63) *
**HbA1c; %**	0.67 (0.29; 1.04) *	0.37 (−0.26; 1.00)	0.88 (0.43; 1.33) *
**Systolic Blood Pressure; mmHg**	0.02 (0.01; 0.04) *	0.02 (−0.01; 0.06)	0.02 (−0.01; 0.04)
**Diastolic Blood Pressure; mmHg**	0.04 (0.01; 0.07) *	0.03 (−0.02; 0.08)	0.02 (−0.01; 0.06)
**Arterial Hypertension**	1.04 (0.42; 1.66) *	1.14 (−0.03; 2.31) *	0.75 (0.07; 1.44)

* *p* < 0.05; Linear regression models adjusted for study, age, sex, BMI, and smoking (exception: models with smoking status and waist circumference exposures not adjusted for BMI). Results are described using standardized regression coefficient β and 95 % CI. All analyses were weighted for drop-out to the MRI examination.

**Table 3 healthcare-10-02310-t003:** Intraglandular tumors in 68 subjects.

Tumors	N	Median Volume (cm^3^)	Mean (cm^3^)	Standard Deviation	Variance	Kurtosis
Right parotid gland	41	0.161	0.299	0.333	0.111	5.95
Left parotid gland	34	0.214	0.56	1.346	1.811	26.91

**Table 4 healthcare-10-02310-t004:** Comparison of imaging studies describing the volume of the parotid gland and other associations.

	Dos Santos et al. (2020) [35]	Heo et al. (2001) [15]	Li et al. (2014) [13]	Mahne et al. (2007) [12]		Medbery et al. (2000) [14]	Ono et al. (2006) [36]	Saito et al. (2013) [34]	Present Study (2022)
**Imaging (plane, layer thickness)**	CT with CA	CT (5 mm)	CT (axial, 1.25 mm)	MRI, 1.5 T (axial, 5 mm)	PET-CT (axial, 5 mm)	MRI, 1.5 T (7 mm)	MRI, 1.5 T (axial, 7 mm)	MRI, 1.5 T	MRI, 1.5 T (axial, 4 mm)
**N-subjects (m; %m)**	49 (45; 92%)	42 (21; 50%)	240 (120; 50%)	64 (25; 39%)	35 (19; 54%)	16 (n.e.)	28 (23; 82%)	35 (20; 57%)	1725 (654; 38%)
**Study design; specifics**	Clinical, oncology study (pre-& post radiation)	Prospective; No volume estimation, measuring MCSA	Prospective; no exclusion of extra parenchymal structures	Retrospective; oncology setting; 2 separate collectives and modalities		Retrospective; oncology setting	Prospective; exclusion of extra parenchymal structures	Retrospective; clinical setting; exclusion of extra parenchymal structures	Prospective, epidemiological study; exclusion of extra parenchymal structures
**Mean parotid gland volume, (range), <SD>**	29 cm^3^ (11–55) <9.5>	{7.5–8.7 MCSA in cm^2^}	25 cm^3^ m: 16.9–35.1 cm^3^ f: 13.9–34.9 cm^3^	bilat.: m: 55 cm^3^ (20–96) <19> f: 36cm^3^ (13–72) <16>	m: 25 cm^3^ (3–42), <12> f: 17 cm^3^ (11–36), <9>	25.3 (9–54) cm^3^	bilat.: 80.2 cm^3^ (46–120) <SEM 3.3>	5–38 cm^3^	23.9 cm^3^ <8.1> (4–61.8) m: 27.82 cm^3^ <8.23> f: 21.60 cm^3^ <6>
**Inter-** **observer reliability; ICC**	Automatic ROI marking according to thresholds	n.e.	n.e.	n.e.	n.e.	4,8%	n.e.	n.e.	7.9%/16.7%; ICC 0.78/0.72
**Laterality**	r = l	r = l	r = l	r = l	r = l	r = l	r = l	n.e.	r = l
**Association with sex (m > f)**	n.e.	*p* > 0.05	*p* < 0.05	*p* < 0.001	*p* = 0.15	n.e.	n.e.	n.e.	*p* < 0.05
**Association with age (range of age in years)**	n.e.	*p* < 0.05 (21–76)	*p* < 0.01 (25–60)	*p* = 0.06 (13–81)	*p* = 0.3 (10–76)	n.e.	n.e.	*p* < 0.05 for age-dependent fat saturation (0.5–87)	*p* < 0.05 (21–79)
**Association with BMI**	n.e.	*p* < 0.05	n.e.	n.e.	n.e.	n.e.	Body weight: *p* < 0.001	n.e.	*p* < 0.05

Bilat.—Bilateral size (sum of left and right gland volume); CA—Contrast agent; f—Female; m—Male; %m—Percentage of male subjects; MCSA-Maximum cross-sectional area; n.e.—Not evaluated; r = l—Equal volumes of right and left parotid gland; ROI—Region of interest; SD—Standard deviation; SEM- Standard error of the mean.

## Data Availability

Primary data from this analysis and SHIP in general is available from a data repository of the University Medical Center Germany on request for scientific purposes (http://www.fvcm.med.uni-greifswald.de/ (accessed on 1 October 2022)).

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
