# Peer review of "Population Based Average Parotid Gland Volume and Prevalence of Incidental Tumors in T1-MRI"

_healthcare, 2022, doi:10.3390/healthcare10112310_

Round 1
Reviewer 1 Report
This manuscript describes a parotid grand volume analysis study using T1-weighted magnetic resonance imaging data in a population-based fashion, which have significant potential to provide information not only as normative data but also as pathophysiological implication which may be associated with chronic diseases.
 The trial is important for better healthcare. However, unfortunately, I have to comment that the quality of the current version of the manuscript strikes me as a first draft, due to the following:
A
1) The hypothesis is unclear. It should be led logically from the previous evidence.
2) There are not enough sentences formed into paragraphs.
3) There are no subheadings nor topic sentences, just a narrative list of previous reports.
4) There are mixed discussions without a priority of the findings, and
5) There is an inconsistency between the hypothesis and the conclusion.
B
1) The raw data of parotic volumes measured are not clearly shown throughout the manuscript despite the figure showing the interrater variability of parotid volume estimated.
2) The figures should include each raw data plots, which are shown in a light color to justify the linear and nonlinear fitting.
3) Whether the screenshots are suitable as a presentation in the paper should be discussed prior to submission.
I think that the authors could reconsider these points to make the manuscript relevant and show how it can be beneficial for interested readers.
I hope you kindly consider my suggestion. Thank you.
Author Response
Thank you kindly for this very constructive evaluation. We strongly agree, that the structure of this article should emphasize the benefit of this study more clearly. In particular, we have revised the introduction, discussion, and conclusion sections in accordance with your suggestions. For details see the list of your remarks and our remarks on revision below.
A
1) The hypothesis is unclear. It should be led logically from the previous evidence.
The hypothesis has been highlighted more clearly in the introduction and is discussed accordingly later on.
2) There are not enough sentences formed into paragraphs.
Paragraphs in general have been restructured.
3) There are no subheadings nor topic sentences, just a narrative list of previous reports.
Paragraphs in general have been restructured.
4) There are mixed discussions without a priority of the findings, and
5) There is an inconsistency between the hypothesis and the conclusion.
The discussion section includes results from several other studies to confirm the value of our findings. It is organized in the same order in which the hypotheses and results are presented.
Hypothesis and conclusion are now congruent.
B
1) The raw data of parotic volumes measured are not clearly shown throughout the manuscript despite the figure showing the interrater variability of parotid volume estimated.
Raw data are now included in the figures concerning associations of sex, age and BMI with parotid gland volume.
2) The figures should include each raw data plots, which are shown in a light color to justify the linear and nonlinear fitting.
see above
3) Whether the screenshots are suitable as a presentation in the paper should be discussed prior to submission.
Please consider, that screenshots and figures are available in appropriate resolution in a separate data file. We consider the content of the screenshot important so that the method used is more imaginable and tangible for the reader.
Reviewer 2 Report
The introduction is insufficient, and the significance of this study is not clear.
Please export the MRI image from the software with high resolution instead of simply using screenshot.
Please provide a demographic table for the subjects involved in this study.
Please provide the name of software for the statistics.
Please uniform the figures such as font and font size.
Please improve the resolution of all figures.
Please consider combining figure 5 and 6, figure 7 and 8. The single figure was not informative.
Author Response
Thank you kindly for this very helpful evaluation. We strongly agree, that the structure of this article should emphasize the significance of this study more clearly. In particular, we have revised the introduction, discussion, and conclusion sections in accordance with your suggestions.
The introduction is insufficient, and the significance of this study is not clear.
The introduction section has now been revised and the hypothesis has been stated more clearly.
Please export the MRI image from the software with high resolution instead of simply using screenshot.
Please provide a demographic table for the subjects involved in this study.
We now have included data plots concerning age, sex and BMI in figure 5 and 6, such adding a demographic component. We feel, that an additional table on demographic data would draw the focus away from the salivary gland context. Demographic tables concerning the SHIP-cohorts are accessible anytime in [Völzke, 2022, Cohort Profile Update: The Study of Health in Pomerania (SHIP)]
Please provide the name of software for the statistics.
The name oft he software has now been included in the method section.
Please uniform the figures such as font and font size.
Font and font size have been edited and are as uniform as we thin is appropriate for different diagrams.
Please improve the resolution of all figures.
Please kindly acknowlede, that all images and figures are given in a separate data file with appropriate resolution and font sizes.
Please consider combining figure 5 and 6, figure 7 and 8. The single figure was not informative
Figure 5 and 6 have been extended by including raw data plots thus merging them would not improve their informative value.
Round 2
Reviewer 1 Report
Thank you for your consideration based on my comments.
However, I think that the quality of the revised manuscript remains immature yet. It may confuse the interested readers.
Therefore, unfortunately, I have to tell you a negative recommendation.
I am sorry for your inconvenience and I hope this manuscript will be submitted to another journal, which will make this research beneficial for interested readers.

Author Response
Dear Editor and Reviewers,
thank you very much for coconsidering our work for publication in Healthcare. Please find below our reply to the issues brought forward in the current round of revision. As suggested, we have performed a major revision leading to a changed manuscript in all parts.
Thank you very much for your valued opinion. Your comments were extremely helpful in restructuring the article. Analogous to your comments in the manuscript-pdf we have made extensive revisions.
- Introduction and conclusion were revised: Subheadings were included and hypothesis highlighted and clearly formulated.
- Paragraphs were put into order and are less fractured.
- The presentation was fixed and adapted according to your valued comments. The histograms of parotid gland volumes left and right side are now merged.
- An additional table for better comparison of existing studies compared to the present study was implemented for the interested reader to better understand the novelty of our findings and to summarize our thorough literature review.
- The conclusion is now structured and picks up the hypothesis of the introduction.
We would like to express our gratitude to all reviewer and the editor for their valuable remarks, which certainly have led to an improvement of the article. We are looking forward to your assessment of the revised manuscript.
Best regards,
Tina Brzoska on behalf of the author team
Reviewer 2 Report
The sentences of content should be structured not like one sentence was one paragraph.
The resolution of all figures definitely should be improved to 300 ppi at leaset.
Please add the demographics table and logically organize the paragraphs instead of one sentence to be a paragraph, then the paper can be accepted. Thank you.
Author Response
Dear Editor and Reviewers,
thank you very much for coconsidering our work for publication in Healthcare. Please find below our reply to the issues brought forward in the current round of revision. As suggested, we have performed a major revision leading to a changed manuscript in all parts.
We very much appreciate your comments and implemented your suggestions accordingly.
- The sentences of content should be structured not like one sentence was one paragraph.
Sentencens were structured and put into paragraphs that now make sense in terms of content.
- The resolution of all figures definitely should be improved to 300 ppi at leaset.
The figures were revised. All of them are now available with 300 ppi.
- Please add the demographics table and logically organize the paragraphs instead of one sentence to be a paragraph, then the paper can be accepted. Thank you.
We added a demographic table (Table 1) comparing in- and excluded subjects. Throughout the manuscript, paragraphs were re-organized.
We would like to express our gratitude to all reviewer and the editor for their valuable remarks, which certainly have led to an improvement of the article. We are looking forward to your assessment of the revised manuscript.
Best regards,
Tina Brzoska on behalf of the author team
Round 3
Reviewer 1 Report
I think that the current version has been sufficiently improved to warrant publication. I hope that this paper will provide valuable information for interested readers.
Author Response
Dear Reviewer,
we are very grateful for your re-evaluation of our manuscript and your approval.
Kind regards,
Tina Brzoska (on behalf of the author team)
Reviewer 2 Report
The authors has given comprehensive response, and the reviewer agreed to accept it.
Author Response

(The authors gave the same response as above.)
